# Lack of airway submucosal glands impairs respiratory host defenses

Lynda S Ostedgaard[1]*, Margaret P Price[1], Kristin M Whitworth[2],
Mahmoud H Abou Alaiwa[1], Anthony J Fischer[3], Akshaya Warrier[1],
Melissa Samuel[2], Lee D Spate[2], Patrick D Allen[1], Brieanna M Hilkin[1],
Guillermo S Romano Ibarra[1], Miguel E Ortiz Bezara[1], Brian J Goodell[1],
Steven E Mather[1], Linda S Powers[1], Mallory R Stroik[1], Nicholas D Gansemer[1],
Camilla E Hippee[1], Keyan Zarei[1,4], J Adam Goeken[5], Thomas R Businga[5],
Eric A Hoffman[4,6], David K Meyerholz[5], Randall S Prather[2], David A Stoltz[1,4,7]*,
Michael J Welsh[1,7,8]*

[1]Department of Internal Medicine and Pappajohn Biomedical Institute Roy J. and Lucille A. Carver College of Medicine, University of Iowa, Iowa City, United States; [2]Division of Animal Science, University of Missouri, Columbia, United States; [3]Department of Pediatrics, Roy J. and Lucille A. Carver College of Medicine, University of Iowa, Iowa City, United States; [4]Department of Biomedical Engineering, University of Iowa, Iowa City, United States; [5]Department of Pathology, Roy J. and Lucille A. Carver College of Medicine, University of Iowa, Iowa City, United States; [6]Department of Radiology, Roy J. and Lucille A. Carver College of Medicine, University of Iowa, Iowa City, United States; [7]Department of Molecular Physiology and Biophysics, Roy J. and Lucille A. Carver College of Medicine, University of Iowa, Iowa City, United States; [8]Howard Hughes Medical Institute, University of Iowa, Iowa City, United States

**\*For correspondence:**
lynda-ostedgaard@uiowa.edu (LSO);
david-stoltz@uiowa.edu (DAS);
michael-welsh@uiowa.edu (MJW)

**Competing interests:** The authors declare that no competing interests exist.

**Abstract** Submucosal glands (SMGs) are a prominent structure that lines human cartilaginous airways. Although it has been assumed that SMGs contribute to respiratory defense, that hypothesis has gone without a direct test. Therefore, we studied pigs, which have lungs like humans, and disrupted the gene for ectodysplasin (*EDA-KO*), which initiates SMG development. *EDA-KO* pigs lacked SMGs throughout the airways. Their airway surface liquid had a reduced ability to kill bacteria, consistent with SMG production of antimicrobials. In wild-type pigs, SMGs secrete mucus that emerges onto the airway surface as strands. Lack of SMGs and mucus strands disrupted mucociliary transport in *EDA-KO* pigs. Consequently, *EDA-KO* pigs failed to eradicate a bacterial challenge in lung regions normally populated by SMGs. These in vivo and ex vivo results indicate that SMGs are required for normal antimicrobial activity and mucociliary transport, two key host defenses that protect the lung.

## Introduction

A thin layer of airway surface liquid (ASL) is the initial point of contact when inhalation and aspiration carry potential pathogens into the lung. ASL serves an important protective function against infection by killing microorganisms with secreted antimicrobial peptides/proteins and by facilitating their removal with mucins and mucociliary transport (MCT) (*Widdicombe and Wine, 2015*; *Wine and Joo, 2004*; *Ganz, 2002*; *Whitsett, 2018*; *Knowles and Boucher, 2002*). ASL is comprised of secretions from two sources: surface epithelia lining airways and submucosal glands (SMGs) in the underlying submucosa. Both produce antimicrobials and mucins (*Widdicombe and Wine, 2015*; *Wine and*

*Joo, 2004*; *Ganz, 2002*; *Whitsett, 2018*; *Knowles and Boucher, 2002*; *Basbaum et al., 1990*; *Fahy and Dickey, 2010*). Humans have abundant SMGs in their cartilaginous airways extending 8–10 airway generations to diameters of 1–2 mm (*Widdicombe and Wine, 2015*; *Choi et al., 2000*; *Whimster, 1986*). Based on their secretory products and multiple previous studies, it has been assumed that SMGs play an important role in respiratory host defense (*Widdicombe and Wine, 2015*; *Wine and Joo, 2004*; *Whitsett, 2018*; *Basbaum et al., 1990*; *Fahy and Dickey, 2010*; *Joo et al., 2015*; *Dajani et al., 2005*; *Bartlett et al., 2013*; *Fischer et al., 2009*). But that assumption has gone without a direct in vivo test, and it has not been possible to determine the separate contributions of SMGs and surface epithelia to airway defense.

Questions about the role of SMGs in protecting the lung have also arisen related to disease. An example is cystic fibrosis (CF), a disease in which respiratory host defense defects cause airway infection (*Quinton, 1999*; *Stoltz et al., 2015*; *Quinton, 2008*; *Rowe et al., 2005*). CFTR is expressed in surface epithelia of proximal airways, SMG of cartilaginous airways, and surface epithelia of non-cartilaginous distal airways; loss of CFTR likely impairs host defense at each of these sites. However, it has been said that loss of CFTR predominantly affects distal airways and/or surface epithelia, implying that defects in SMGs may have little role as a primary contributor to CF lung disease (*Tiddens et al., 2010*; *Oppenheimer and Esterly, 1975*; *Boucher, 2019*; *Thelin and Boucher, 2007*; *Ratjen, 2012*).

Although numerous studies have investigated the role of surface epithelia in respiratory host defense, far fewer have addressed the role of SMGs in airway defense. Several factors have limited investigation and knowledge. Serous cells have been isolated from SMG and their electrolyte transport properties elucidated (*Lee and Foskett, 2010a*; *Lee and Foskett, 2010b*). SMG cells have also been grown as epithelia; however, their differentiation has not yet replicated that of in vivo SMGs, and they have not been widely used (*Finkbeiner et al., 2011*; *Fischer et al., 2010*; *Widdicombe et al., 2012*). Yet, a limitation of both cultured and isolated SMG cell models is that they do not recreate the SMG architecture, and we are not aware of their use to assess host defenses. Mice are the animals most commonly used to study lung function. However, they lack SMGs except in the most proximal part of the airway, whereas humans have abundant SMGs in cartilaginous airways (*Widdicombe and Wine, 2015*; *Choi et al., 2000*; *Meyerholz et al., 2018a*; *Borthwick et al., 1999*). Moreover, mice have not proven to be a good model for human diseases that involve SMGs, such as cystic fibrosis (CF) (*Grubb and Boucher, 1999*; *Guilbault et al., 2007*). Use of SMGs isolated from human samples is limited due to their inadequate availability and uncertainty about changes that may have occurred due to disease-related remodeling.

Despite these limitations, much work suggests that SMGs play an important role in respiratory host defense. Previous studies indicate that SMGs produce multiple different antimicrobials (*Widdicombe and Wine, 2015*; *Ganz, 2002*; *Basbaum et al., 1990*; *Joo et al., 2015*; *Dajani et al., 2005*; *Bartlett et al., 2013*; *Fischer et al., 2009*). A diverse complement of antimicrobial peptides/proteins may be particularly important because antimicrobials exhibit synergistic activity against bacteria (*Singh et al., 2000*; *Abou Alaiwa et al., 2014*). A study of ferret trachea xenografts, which contained SMGs, transplanted into immune mice indicated that it produced more antimicrobials and had greater antimicrobial activity than ferret cells seeded on denuded rat xenografts, which did not have SMGs (*Dajani et al., 2005*). Evidence suggests that SMGs may also contribute to MCT. Previous studies suggest that interventions that stimulate SMG secretion increase MCT (*Widdicombe and Wine, 2015*; *Wanner et al., 1996*). For example, treating excised pig and ferret tracheas with acetylcholine and other agonists increased MCT, and inhibiting liquid secretion attenuated the increase (*Ballard et al., 2002*; *Jeong et al., 2014*). However, uncertainty remains about the contribution to MCT and antimicrobial activity by SMGs vs. surface epithelia, the role of SMGs under basal vs. stimulated conditions, and the contribution of SMGs in vivo vs. ex vivo.

To investigate the role of SMGs in respiratory host defense, we chose to study pigs. Similarity between pig and human lungs regarding the type, amount, and distribution of airway epithelial cells and SMGs; airway and lung size; airway antimicrobials; and airway transepithelial electrolyte transport make them ideal for our studies (*Rogers et al., 2008*; *Prather et al., 2013*; *Judge et al., 2014*). SMGs from pigs have also been used as a model in many physiological studies (*Lee and Foskett, 2010a*; *Lee and Foskett, 2010b*; *Ballard et al., 2002*; *Ballard et al., 1995*; *Joo et al., 2010*; *Joo et al., 2002*). In addition, because SMGs are present at birth, pigs can be studied as newborns, thereby avoiding secondary consequences of disease. Genetically modified pigs also provide a good

model for airway disease such as CF, where they have highlighted the importance of antimicrobial peptides/proteins and MCT for host defense (*Hoegger et al., 2014*; *Stoltz et al., 2010*; *Pezzulo et al., 2012*; *Shah et al., 2016*; *Ermund et al., 2018*).

Formation of glands and other epithelial appendages like hair and teeth is initiated by the ecto-dysplasin pathway (*Rawlins and Hogan, 2005*; *Laurikkala et al., 2002*; *Mikkola, 2009*; *Jaskoll et al., 2003*). This pathway includes the soluble ligand ectodysplasin A (EDA), the EDA receptor EDAR, and the EDA transducer EDARADD. Mutations in each of the genes in this pathway have been reported in humans who develop the disease hypohidrotic ectodermal dysplasia (HED) (*Pääkkönen et al., 2001*; *Kere et al., 1996*; *Cluzeau et al., 2011*; *Schneider et al., 2001*). HED defects include loss or reduction in glands, hair, and teeth (*Dietz et al., 2013*; *Clarke et al., 1987*; *Capitanio et al., 1968*; *Reed et al., 1970*). There are also variable reports of recurrent respiratory tract infections, asthma-like symptoms, and otitis media (*Dietz et al., 2013*; *Clarke et al., 1987*; *Reed et al., 1970*; *Callea et al., 2013*; *Beahrs et al., 1971*). *EDA* mutations in mice, dogs, and cattle cause loss of nasal, submandibular, and submucosal glands, and there are limited reports investigating respiratory symptoms and mucus accumulation (*Jaskoll et al., 2003*; *Azar et al., 2016*; *Casal et al., 2005a*; *Seeliger et al., 2005*; *Vasiliadis et al., 2019*). Reports that *EDA* mutations can cause a decrease or loss of SMGs and produce lung abnormalities suggest the feasibility of using *EDA* gene disruptions to develop an animal model without SMGs. However, the type and severity of *EDA* mutations (loss or reduced function) is not always clear, the respiratory consequences have been little investigated, and we lack knowledge about how *EDA* disruption affects host defenses.

We hypothesized that disrupting the *EDA* gene in pigs would eliminate SMGs and thereby impair two key respiratory host defenses, bacterial killing and MCT.

## Results

### Gene editing generated *EDA-KO* pigs

*EDA* encodes a membrane protein that when cleaved releases a secreted protein fragment containing a collagen motif and a TNF-like ligand (*Rawlins and Hogan, 2005*; *Schneider et al., 2001*; *Sadier et al., 2014*; *Figure 1A and B*). Trimerization of the collagen domain is a prerequisite for trimerization of the C-terminal TNF motif that then binds to the receptor EDAR inducing signaling through EDARADD, which is required for proper gland development (*Schneider et al., 2001*; *Swee et al., 2009*). Mutations in the collagen domain prevent trimerization and subsequent binding of the TNF domain to the EDAR (*Pääkkönen et al., 2001*; *Schneider et al., 2001*).

To generate pigs lacking SMG, we targeted exon 4 of the *EDA* gene; exon 4 encodes the collagen domain (*Figure 1B*). We used CRISPR/Cas9 gene editing, injecting the guide RNAs and *Cas9* mRNA into porcine zygotes (*Yuan et al., 2017*; *Chen et al., 2018*; *Redel et al., 2019*). Blastocysts were then implanted into surrogate sows. We generated four litters, and in all the piglets, we found indels in exon 4 that predicted a loss of function. *Figure 1C* shows an example of PCR fragments from six piglets in one litter suggesting large sequence deletions in most piglets. We sequenced DNA from all animals, which revealed insertions, deletions, and in some cases, nonsense mutations that either disrupted the collagen domain or truncated the protein (*Figure 1—figure supplement 1*). For simplicity, we refer to all these pigs as *EDA-KO*.

### The appearance of *EDA-KO* piglets differed from that of wild-type piglets

The numbers of male and female piglets were approximately equal. *EDA-KO* pigs had birth weights (1.48 ± 0.48 kg, mean ± SD, n = 20) similar to those of wild-type controls (1.54 ± 0.31 kg, n = 18) (p=0.94). Of note, the *EDA* gene is located on the X chromosome. However, on initial exam, we could discern no sex differences in *EDA-KO* piglets. The only observed difference in behavior of *EDA-KO* piglets was sneezing; sneezing began within hours of birth and persisted.

The *EDA-KO* piglets all appeared healthy. However, there were obvious physical differences between *EDA-KO* and wild-type pigs at birth (*Figure 2A*). *EDA-KO* pigs had sparse hair on the back distributed as linear patterns of hairy stripes alternating with hairless skin (*Figure 2B*, *Figure 3—figure supplement 1*). They had a bald circular patch on the top of their heads (*Figure 2C*). *EDA-KO* piglets also lacked hair on their upper and lower eyelids but retained eyelashes (*Figure 2D*). These

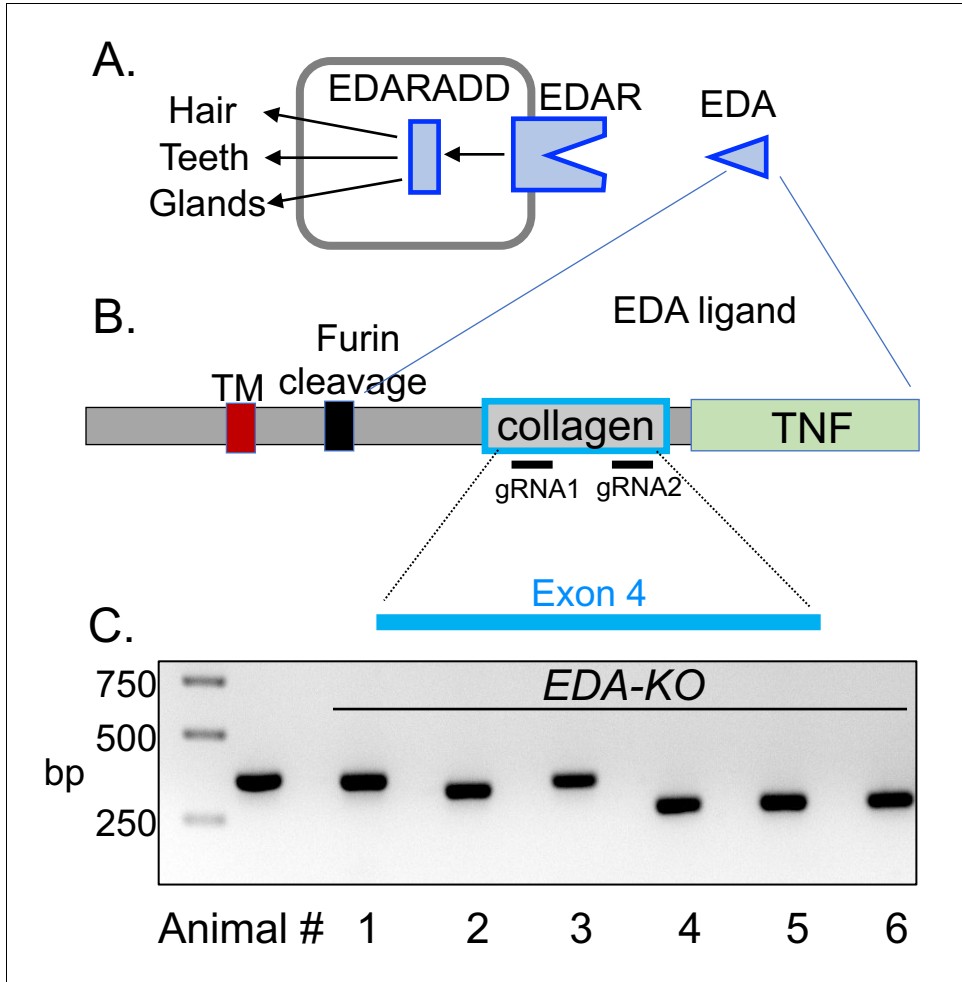

**Figure 1.** CRISPR/Cas9 editing produced pigs with a disrupted *EDA* gene (*EDA-KO*). (**A**) Diagram of interaction of EDA, EDAR, and EDARADD at the epithelial placode. (**B**) Schematic of EDA protein domains, including the transmembrane domain (TM), furin cleavage site, collagen domain, and TNF domain. Exon 4 of the *EDA* gene encodes the collagen domain. Relative positions of guide RNAs are shown. (**C**) PCR fragments from six edited pigs from one litter. Lane two shows position of predicted wild-type product. Pigs 1–5 were male; pig six was female. The online version of this article includes the following figure supplement(s) for figure 1:

**Figure supplement 1.** Nucleotide sequence of exon 4 of *EDA* gene and sequences of edited pigs.

body hair phenotypes are similar to those reported for other animals with mutations in the ectodysplasin pathway (*Vasiliadis et al., 2019*; *Casal et al., 2005b*).

To minimize potential consequences of secondary inflammation and/or infection that might occur as *EDA-KO* piglets age, we studied piglets within 2–8 days of birth.

### *EDA-KO* piglets lacked airway SMGs

We detected and quantified SMGs histologically. The conducting airways of wild-type pigs had SMGs (*Figure 3A and B*). As is observed in humans (*Widdicombe and Wine, 2015*; *Choi et al., 2000*; *Whimster, 1986*; *Meyerholz et al., 2018a*), the numbers of SMGs decreased from proximal trachea to bronchi and bronchioles. In contrast, airways of *EDA-KO* piglets lacked SMGs. Consistent with loss of airway SMGs, *EDA-KO* pigs lacked glands in other tissues (*Figure 3—figure supplement 1*).

Mucins MUC5AC and MUC5B are expressed in airway surface epithelia of humans and pigs (*Fahy and Dickey, 2010*; *Ostedgaard et al., 2017*; *Ermund et al., 2017*; *Okuda et al., 2019*). Immunostaining revealed no difference in the localization of MUC5AC or MUC5B in surface epithelia

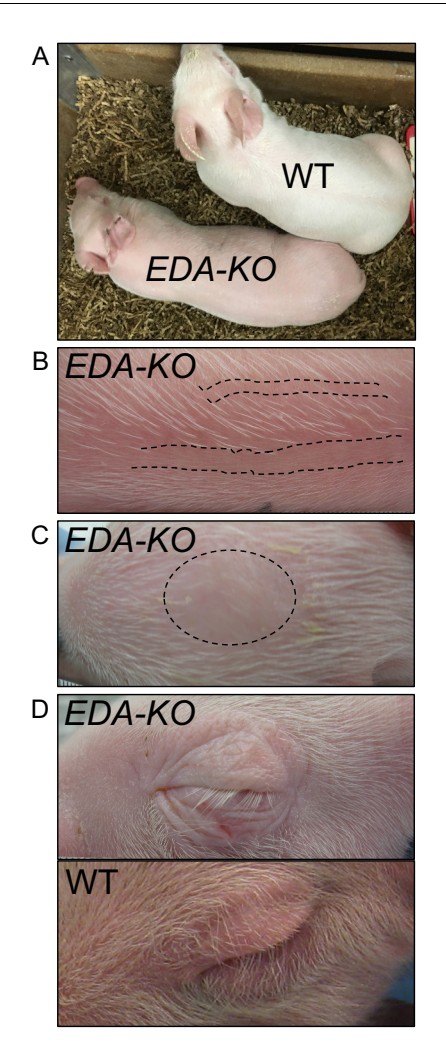

**Figure 2.** *EDA-KO* piglets had an appearance consistent with hypohidrotic ectodermal dysplasia. (**A**) Physical appearance of wild-type (WT) and *EDA-KO* piglets. (**B**) Image of skin on the back of an *EDA-KO* pig showing alternating bands of hair and bare skin (dotted lines). (**C**) Image of skin from the top of head of *EDA-KO* showing bald spot. (**D**) Eyelids in *EDA-KO* pig lacked hair, although eyelashes were intact. Eyelids of a wild-type pig are shown for comparison.

of *EDA-KO* and wild-type airways (*Figure 4A*). The ciliated marker β-tubulin IV also showed similar immunostaining patterns in *EDA-KO* and wild-type piglets (*Figure 4B*).

Lack of SMGs throughout the lung's entire conducting airways is consistent with disruption of the *EDA* gene, which directs initiation of gland duct formation. These results indicated that *EDA-KO* piglets can serve as a model for in vivo testing of the hypothesis that lack of SMGs impairs respiratory host defenses.

## The ASL of *EDA-KO* piglets had a reduced capacity to kill *S. aureus*

It is well known that SMG cells produce a variety of antimicrobials, including lysozyme and lactoferrin (*Basbaum et al., 1990*; *Joo et al., 2015*; *Dajani et al., 2005*; *Fischer et al., 2009*; *Widdicombe et al., 2012*). Thus, we expected that loss of SMGs in *EDA-KO* piglets would decrease bacterial killing by ASL. To specifically test bacterial killing activity in ASL, we used an assay that we had previously developed (*Pezzulo et al., 2012*; *Shah et al., 2016*). We labeled *Staphylococcus aureus* with biotin and linked them to streptavidin-coated gold grids (*Figure 5A*). We introduced a small tracheal window in the newborn pigs and briefly (1 min) placed the bacteria-coated grid on the

trachea surface. We then assessed ASL bactericidal activity using a Live/Dead assay to quantify the % of dead bacteria. This assay measures antibacterial activity in vivo and has the advantages that it is not affected by MCT, phagocytes, variable bacteria delivery or recovery, or bacterial multiplication (*Pezzulo et al., 2012*).

By 1 min, ASL in wild-type piglets killed approximately half of the bacteria (*Figure 5B and C*). These data are similar to bacterial killing in previous studies of wild-type pigs (*Pezzulo et al., 2012*; *Shah et al., 2016*). In contrast, ASL in *EDA-KO* piglets killed less than one-quarter of the bacteria. These results indicate that loss of SMGs impairs antimicrobial activity on the proximal airway surface. These data are consistent with earlier findings that SMGs produce abundant antimicrobials (*Basbaum et al., 1990*; *Joo et al., 2015*; *Dajani et al., 2005*; *Fischer et al., 2009*).

### *EDA-KO* piglets lacked mucus strands and had decreased MCT ex vivo

Acinar cells in SMGs of pigs and humans produce MUC5B, a secreted gel-forming mucin that is the major structural protein of SMG mucus (*Ostedgaard et al., 2017*; *Ermund et al., 2017*; *Wu et al., 2007*; *Thornton et al., 2018*). In pigs and humans, mucus emerges from SMG ducts onto the airway surface in the form of strands (*Hoegger et al., 2014*; *Ermund et al., 2018*; *Ostedgaard et al., 2017*; *Ermund et al., 2017*; *Fischer et al., 2019*; *Tipirneni et al., 2018*; *Trillo-Muyo et al., 2018*; *Xie et al., 2020*). Mucus strands sweep across the airway surface propelled by cilia and bind particulate material and bacteria. Thus, they remove potentially injurious material from the lungs. Although airway surface epithelia of *EDA-KO* pigs expressed mucins, the absence of SMGs suggested that *EDA-KO* pigs would lack mucus strands.

To test this prediction, we removed trachea from newborn pigs, submerged the tracheal segments in saline, added fluorescent nanospheres to the saline to label the mucus, stimulated SMG secretion with methacholine, and used confocal microscopy to watch mucus strands sweep across the airway surface, as previously described (*Hoegger et al., 2014*; *Fischer et al., 2019*). To quantify mucus strands, we measured the number of strands crossing a predefined field. During a 15-min observation period, hundreds of mucus strands crossed the field in wild-type piglet airways (*Figure 6A and B*, *Video 1*). These results are similar to previous findings (*Fischer et al., 2019*). In contrast, *EDA-KO* airways had no or trivial numbers of mucus strands (*Figure 6A and B*, *Video 2*). Ciliary beating was similar in wild-type and *EDA-KO* piglets, indicating that the lack of mucus strands was not due to reduced ciliary activity (*Figure 6C*).

Lack of mucus strands suggested that MCT would be impaired. To test this prediction, we used an approach we previously developed, applying 500 μm metallic spheres to the tracheal surface and tracking their movement (*Fischer et al., 2019*). This method allowed us to assess MCT and the role of mucus that attached to the spheres. After treating tracheal segments with methacholine to stimulate SMG secretion, we placed spheres on the airway and followed their movement with time-lapse photography. In wild-type airways, mucus strands sweeping over the surface often wrapped around the spheres and pulled them off to the edge of the tracheal segment, as we previously reported (*Figure 6D and E*, *Video 3*; *Fischer et al., 2019*). However, without mucus strands in *EDA-KO* airways to initiate movement, spheres less frequently moved (*Figure 6D and E*, *Video 4*). Bits of mucus attached to the spheres, but instead of clearing spheres to the edge of the airway segment, the spheres spun in place due to the action of the underlying cilia.

These ex vivo results emphasize the importance of SMGs in facilitating MCT by producing mucus and assembling it into strands.

### *EDA-KO* piglets had impaired MCT in vivo

Based on our ex vivo results, we hypothesized that loss of SMGs in *EDA-KO* piglets would disrupt MCT in vivo. We measured MCT in spontaneously breathing, non-intubated, sedated pigs using methods previously described (*Hoegger et al., 2014*; *Fischer et al., 2019*). We insufflated radio-dense tantalum microdisks (350 μm), obtained a high-resolution CT scan every 9 s for 6.3 min (total 44 scans), and tracked the position of individual microdisks. We measured MCT before and after stimulating SMG secretion with methacholine.

Under basal conditions, individual microdisks followed a trajectory toward the larynx in wild-type piglets, consistent with earlier results (*Figure 7A*, *Video 5*). In contrast, microdisks showed little movement in *EDA-KO* piglets (*Figure 7A*, *Video 6*). This difference was reflected in a decreased

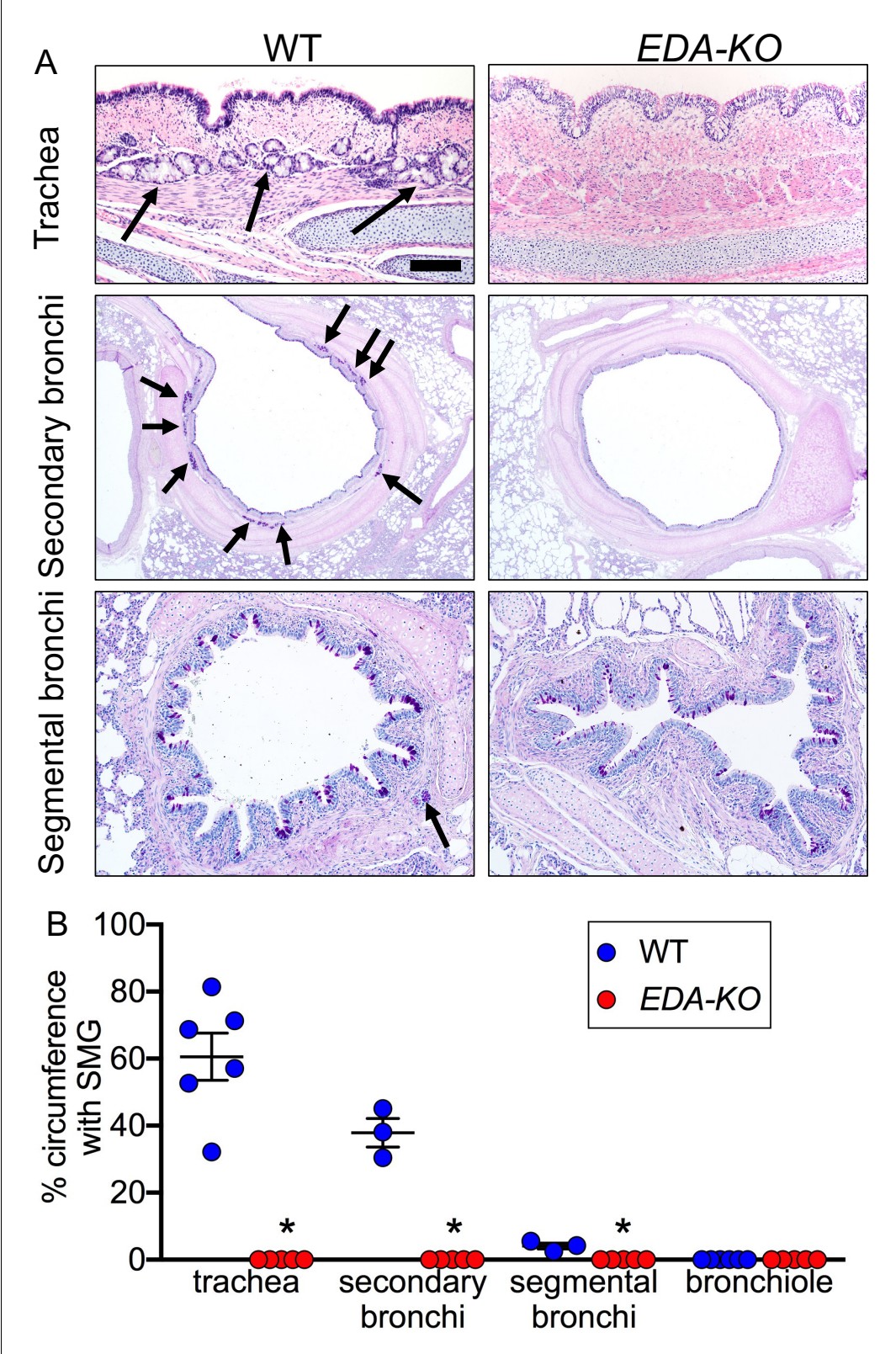

**Figure 3.** *EDA-KO* pigs lacked SMGs. (**A**) Sections of wild-type and *EDA-KO* conducting airways, HE (top) and dPAS (middle and bottom) stains. Arrows point to SMGs. Scale bar = 125 mm (top and bottom panels) and 625 mm (middle panels). (**B**) Data are percentage of airway circumference containing SMG in indicated parts of the airway. Each data point is from a different pig. Trachea: n = 6 wild-type and 5 *EDA-KO*, *p=0.004. Secondary

*Figure 3 continued on next page*

*Figure 3 continued*

bronchi: n = 3 wild-type and 5 *EDA-KO*, p=0.018. Segmental bronchi: n = 3 wild-type and 5 *EDA-KO*, p=0.018. Bronchioles: n = 6 wild-type and 5 *EDA-KO*, p=1.00. Statistical analysis was by Mann-Whitney test.
The online version of this article includes the following figure supplement(s) for figure 3:

**Figure supplement 1.** Histopathology of nasal mucosa, nasal planum, and skin of wild-type and *EDA-KO* piglets.

percentage of time that microdisks were in motion in *EDA-KO* pigs (*Figure 7B*). The speed of disks that did move was similar in both wild-type and *EDA-KO* piglets (*Figure 7C*). These changes were associated with a non-statistically significant trend toward decreased clearance of microdisks from the lungs of *EDA-KO* piglets under basal conditions (*Figure 7D*).

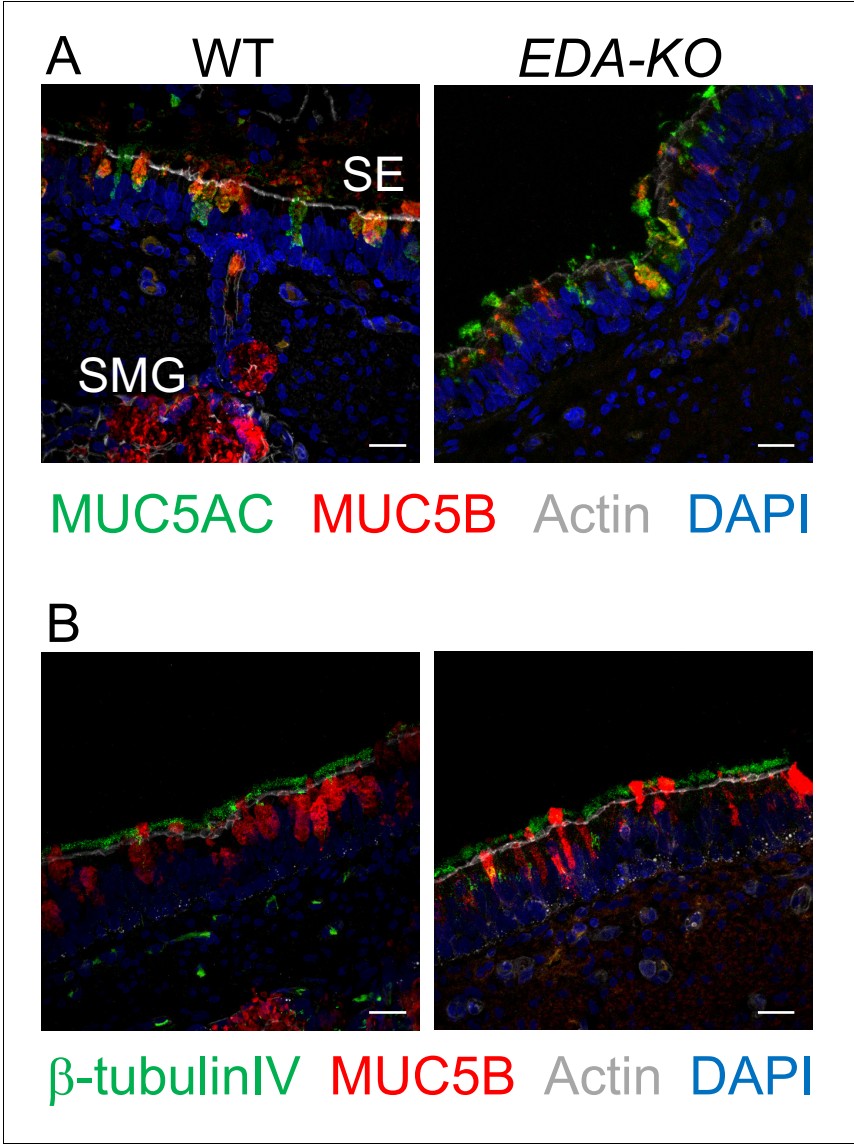

**Figure 4.** MUC5AC, MUC5B, and β-tubulin IV immunolocalization is similar in airway surface epithelia of wild-type and *EDA-KO* piglets. Images are confocal immunofluorescence of wild-type and *EDA-KO* trachea. In all panels, actin is labeled with phalloidin in grey and nuclei are labeled with DAPI in blue. Scale bar = 20 μm. (**A**) Images show goblet cells expressing MUC5AC (green) and MUC5B (red). A portion of a SMG is indicated; SMGs were detected in wild-type only. (**B**) Images indicate ciliated cells (β-tubulin IV, green) and MUC5B (red).

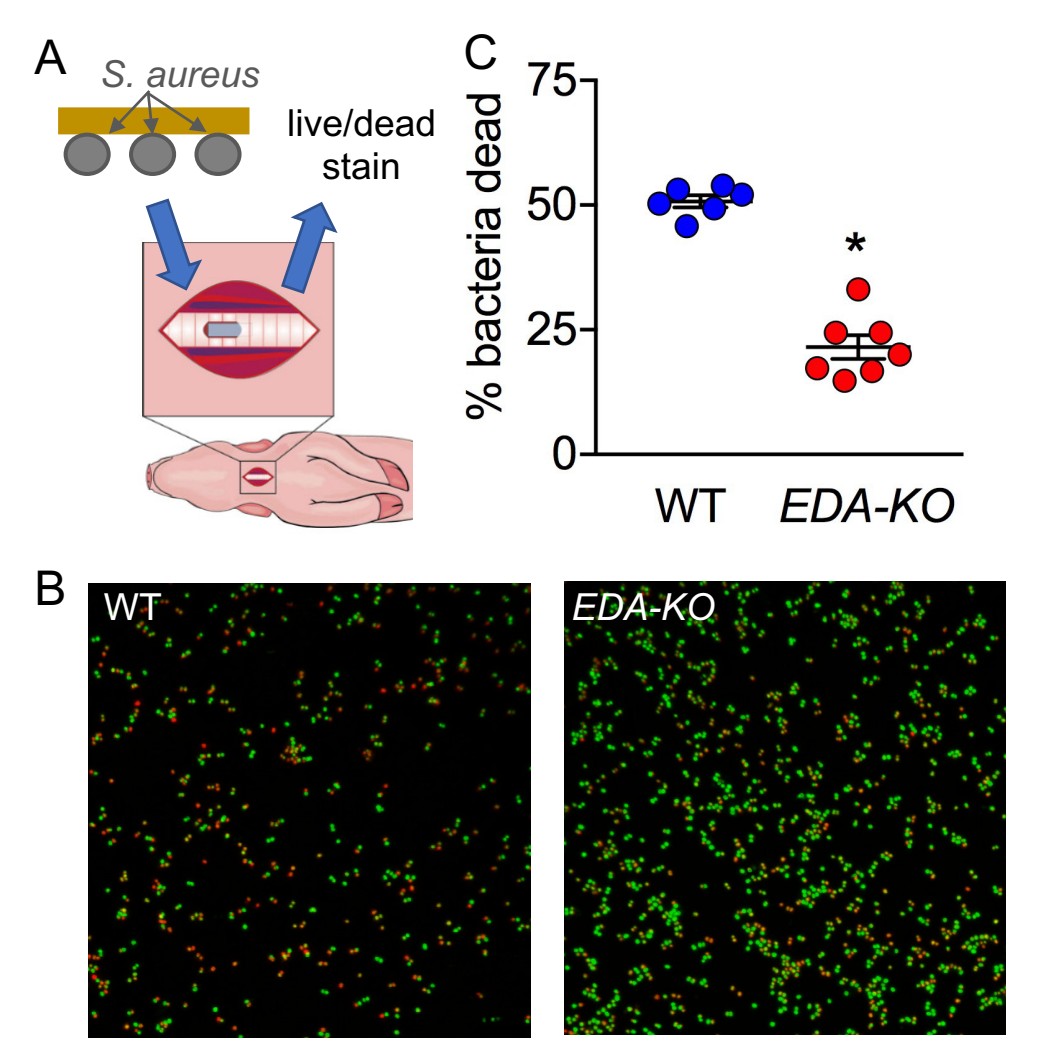

**Figure 5.** ASL of *EDA-KO* pigs has impaired killing of *S.aureus* in vivo. (**A**) Schematic showing *S. aureus* attached to gold grids by biotin-streptavidin linkages. The grids were placed on ASL of trachea for 1 min in vivo. Then the bacteria were counted and the percentage that were dead was determined. (**B**) Example of live (green)/dead (red) staining of bacteria after the grid was removed from the airway. (**C**) Percentage of dead bacteria. *p<0.0012 by Mann-Whitney test.

We also stimulated SMG secretion with the cholinergic agonist methacholine and measured MCT. In wild-type piglets, microdisks followed a trajectory up the large airways and were in motion for most of the time (*Figure 7A and B*, *Video 7*). There was less movement in *EDA-KO* piglets (*Figure 7A and B*, *Video 8*). For microdisks that did move, there was a non-statistically significant trend for a decreased mean speed in *EDA-KO* piglets (*Figure 7C*). In contrast to wild-type piglets, *EDA-KO* piglets failed to clear microdisks from the lung during the course of the study (*Figure 7D*). These findings indicate that loss of SMGs in *EDA-KO* piglets impaired MCT in vivo.

## Airways of *EDA-KO* piglets had an impaired ability to eradicate bacteria

Finding decrements in host defense led us to hypothesize that *EDA-KO* pigs would fail to normally eradicate bacteria. Moreover, we predicted that any defect would be more pronounced in proximal cartilaginous airways, which contain SMGs, than in distal airways. To test this hypothesis, we challenged piglets with a defined inoculum of *S. aureus* delivered by aerosol into the trachea. We used

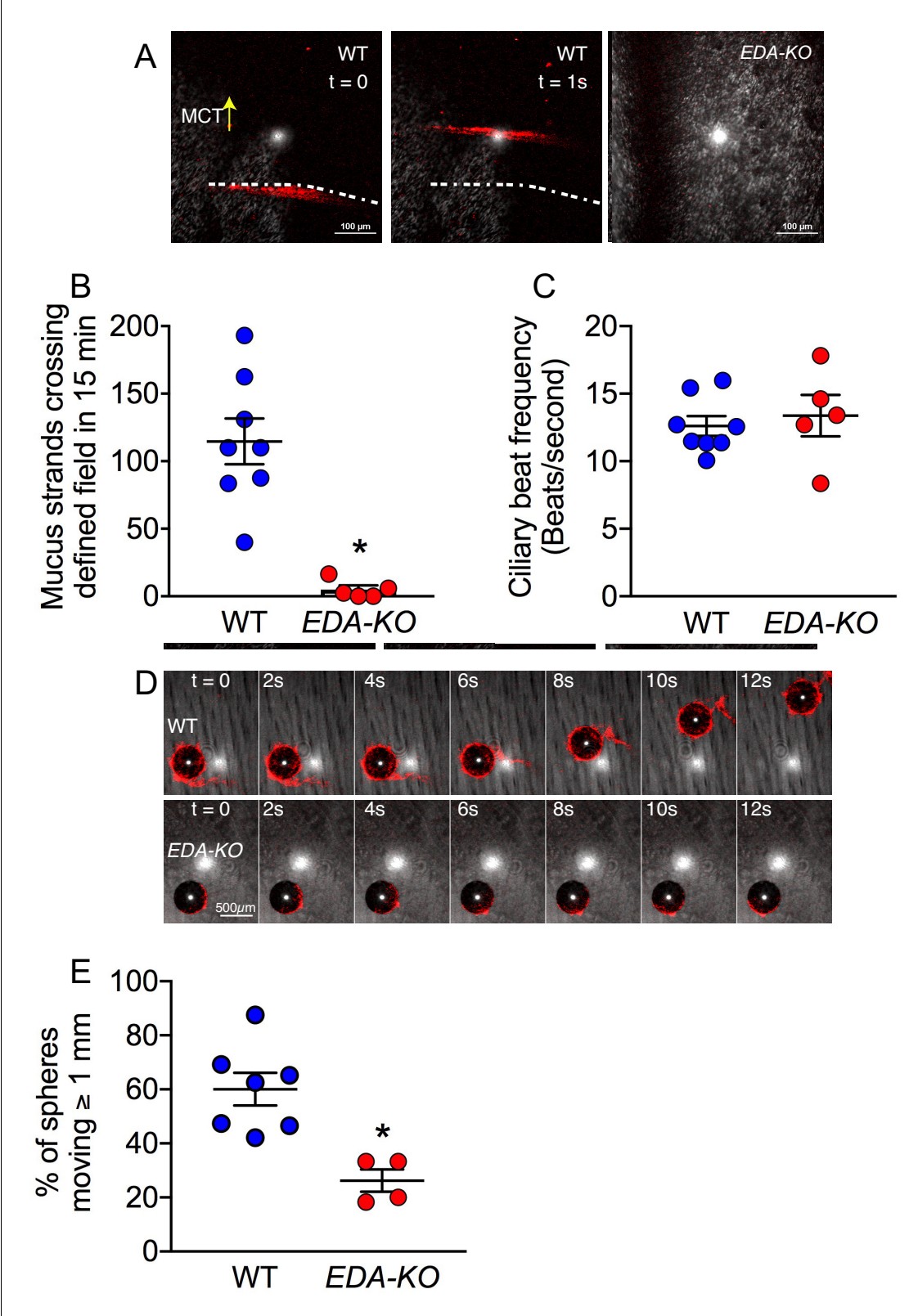

**Figure 6.** Loss of SMGs eliminates mucus strands and impairs MCT ex vivo. (**A**) Data are confocal images of tracheal surface of wild-type and *EDA-KO* pigs. Mucus was labeled with fluorescent nanospheres (red). Left and middle panel show movement of mucus strand with time. Strands were not observed in *EDA-KO* airway in right panel. The white spots in the middle of the field are reflected light. (**B**) Number of mucus strands crossing the microscopy field in 15 min. N = 8 wild-type and 5 *EDA-KO* pigs. *p=0.0016. (**C**) Ciliary beat frequency on trachea under methacholine stimulation. N = 8

*Figure 6 continued on next page*

*Figure 6 continued*

wild-type and 5 *EDA-KO* pigs. p=0.4351. (**D**) Metallic spheres were dropped onto the airway surface and movement was tracked with time. Mucus was labeled with fluorescent nanospheres (red). Images from wild-type airway show a mucus strand attached to a sphere and pulling it across the field. Images from *EDA-KO* airway show mucus attached to sphere that was rolling in place. (**E**) Fraction of metallic spheres that moved at least 1 mm during a 15-min observation period. N = 7 wild-type and 4 *EDA-KO* pigs. * indicates p=0.0061. For panels B, C, and E, each dot represents a different pig. Statistical significance was evaluated with a Mann-Whitney test.

*S. aureus* because it commonly infects humans, and it is frequently found in humans and pigs with CF (*Stoltz et al., 2010*; *Razvi et al., 2009*).

Four hours after aerosolization, we recovered and quantified *S. aureus*. To sample an airway region that has abundant SMGs, we obtained tracheal washes. *EDA-KO* piglets had ~10,000 times more *S. aureus* in tracheal washes than wild-type piglets (*Figure 8*). To sample an airway region that has a mixture of small SMG-containing airways, small airways lacking SMGs, and alveoli, we performed bronchoalveolar lavages. *EDA-KO* piglets had ~100 times more *S. aureus* in BAL liquid than wild-type piglets. To sample predominantly peripheral lung, we homogenized samples of lung. Counts of *S. aureus* did not statistically differ between *EDA-KO* and wild-type piglets, although there was a trend for fewer sterile airways in *EDA-KO* piglets. Although the quantitative delivery to and relative recovery of *S. aureus* in various airway regions is unknown, the data suggest that *EDA-KO* pigs may be less able to eliminate viable bacteria in large airways where SMGs are located than in more distal lung regions that lack SMGs. We speculate that distal airway host defenses might be relatively spared in *EDA-KO* pigs, although similar studies done at varying times after bacteria delivery could be revealing.

## Discussion

These results indicate that airway SMGs protect the lung by contributing to two critical respiratory host defenses, bacterial killing by ASL and production of strands of mucus. As a result, when challenged with *S. aureus*, *EDA-KO* lungs were compromised in their ability to eradicate bacteria, especially in regions normally populated by SMGs. Results from many previous studies underlie the assumption that SMGs play a critical role in host defense. Our findings provide direct evidence for that hypothesis, both in vivo and ex vivo.

Surface airway epithelia can also produce antimicrobials and mucins that facilitate bacterial

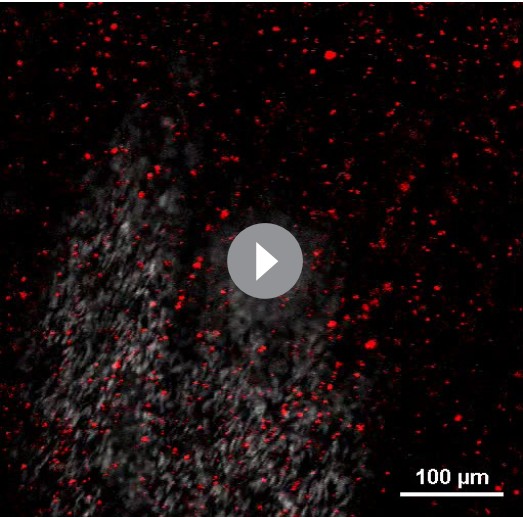

**Video 1.** Mucus strands moving on wild-type trachea stimulated with methacholine. Mucus strands were labeled with fluorescent nanospheres (red). Video is real time. Scale bar 100 μm. White dot in center is from reflected light.

https://elifesciences.org/articles/59653#video1

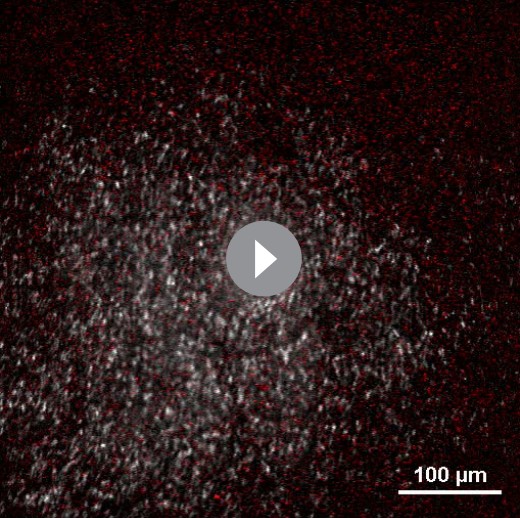

**Video 2.** Surface of *EDA-KO* trachea imaged as in *Video 1*.

https://elifesciences.org/articles/59653#video2

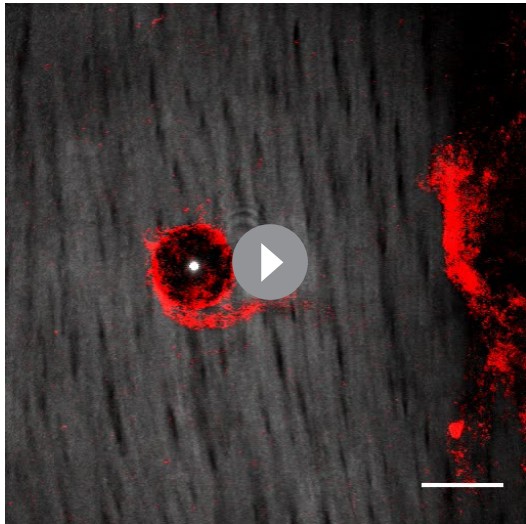

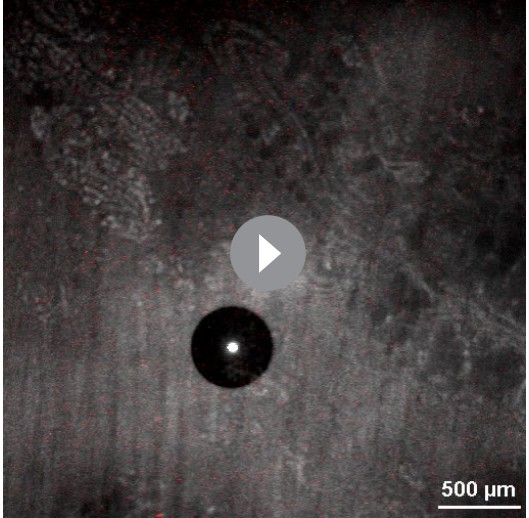

**Video 3.** A 500 µm metallic sphere was placed on a wild-type pig trachea. Mucus strands were labeled with fluorescent nanospheres (red). Mucus strand attaches to and initiates movement of the sphere, pulling it to the edge of the airway segment. Duration of video clip was 30 s and is compressed to 6 s here. Scale bar 500 µm.

https://elifesciences.org/articles/59653#video3

**Video 4.** Imaging as in *Video 3* on *EDA-KO* pig trachea. A sphere is shown spinning backwards and failing to move over the surface of the airway. Mucus attaches to the sphere's surface allowing ready detection of sphere rotation. Duration of video clip was 900 s and is compressed to 32 s.

https://elifesciences.org/articles/59653#video4

killing and MCT (*Widdicombe and Wine, 2015*; *Ganz, 2002*; *Bartlett et al., 2013*; *Fischer et al., 2009*). What then is the evolutionary benefit of having SMGs in humans and pigs? One potential advantage is that SMGs markedly expand the number of epithelial cells available to produce antimicrobials and mucus and deliver them onto the airway surface (*Widdicombe and Wine, 2015*; *Choi et al., 2000*; *Reid, 1960*). In addition, innervation of SMGs by vagal cholinergic efferents enables them to rapidly secrete SMG products on demand (*Wine, 2007*; *Widdicombe, 2002*; *Ballard and Spadafora, 2007*). The ability to quickly deliver copious amounts of antimicrobials, mucins, and other mucus products could be critically important for responding to acute

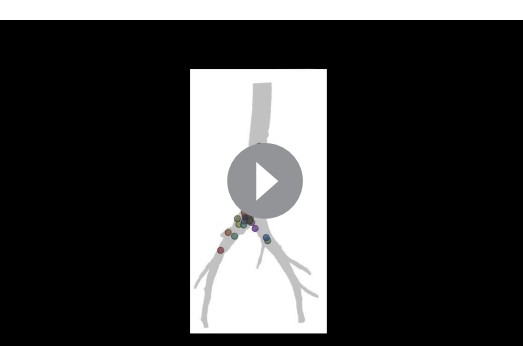

**Video 5.** Transport of microdisks in wild-type pig airways under basal conditions. The head is above the top of the image and tail below the bottom. Microdisks were insufflated into the airways and tracked by sequential CT scanning. Each microdisk is represented by a different colored circle; circles are ~280 times the area of microdisks to aid visualization. When microdisks reach the larynx, they disappear. Video is compressed from original duration of 6.3 min.

https://elifesciences.org/articles/59653#video5

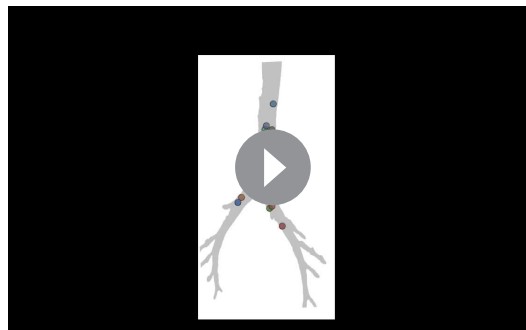

**Video 6.** Transport of microdisks in *EDA-KO* pig airways under basal conditions. Procedures are as described in legend of *Video 5*.

https://elifesciences.org/articles/59653#video6

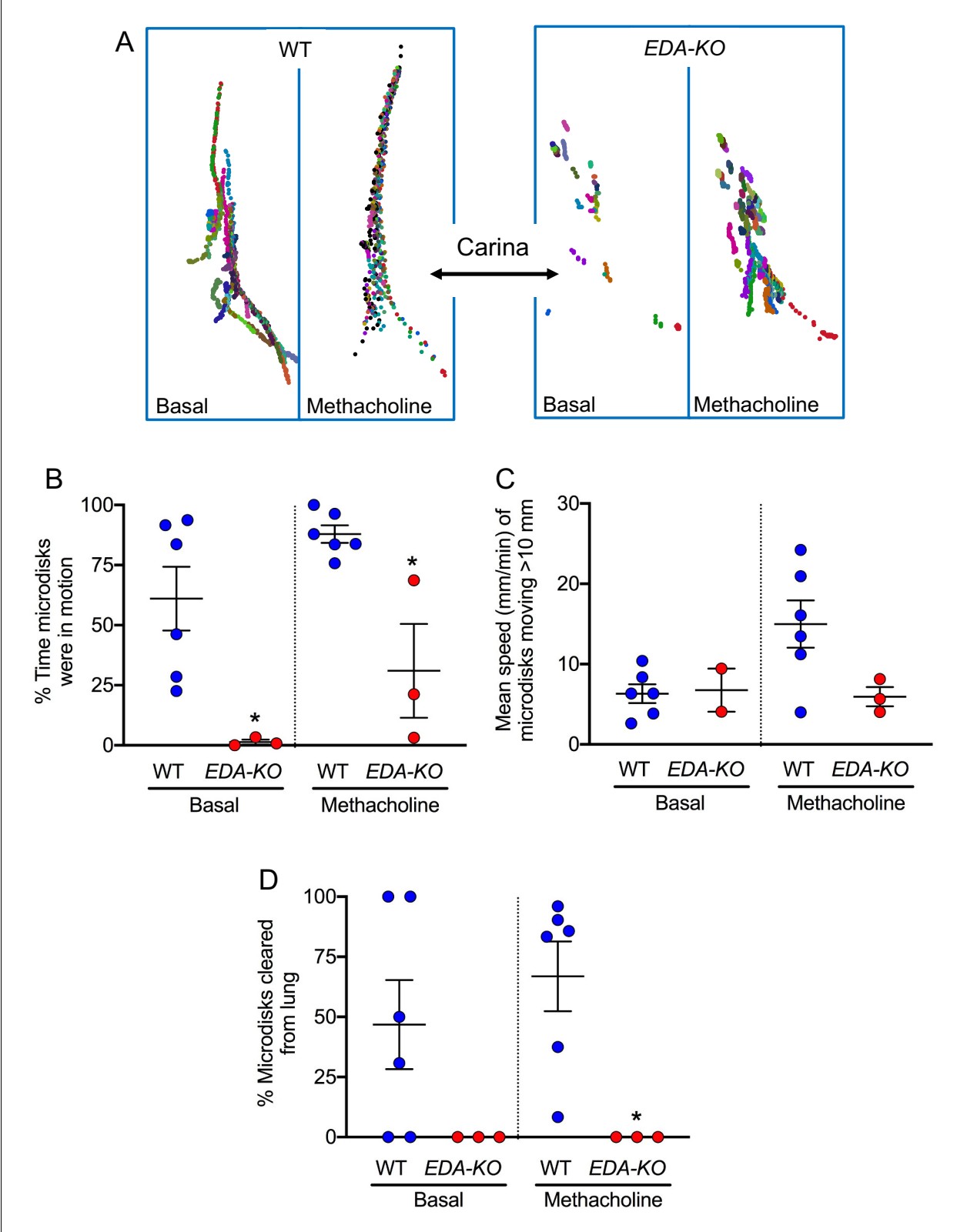

**Figure 7.** *EDA-KO* pigs have impaired MCT in vivo. MCT was assessed by insufflating tantalum microdisks in sedated, spontaneously breathing pigs followed by acquisition of high-resolution CT scans every 9 s for 6.3 min (total 44 scans). Positions of individual microdisks were tracked. Pigs were studied under basal conditions and after stimulating submucosal gland secretion with intravenous methacholine. N = 6 wild-type and 3 *EDA-KO* pigs. Statistical significance between data from wild-type and *EDA-KO* was evaluated with a Mann-Whitney test. (**A**) Examples of individual microdisks

*Figure 7 continued on next page*

*Figure 7 continued*

(different colors) tracked in wild-type (left) and *EDA-KO* (right) pigs. Position of carina is indicated. (B) Percentage of time microdisks were moving. * indicates p=0.0238 under basal conditions and p=0.0238 under methacholine-stimulated conditions. (C) Mean speed (mm/min) of microdisks that moved more than 10 mm. One *EDA-KO* pig had no microdisks moving >10 mm under basal conditions; therefore, only two data points and the range are shown in that case. * indicates p=0.8571 under basal conditions and p=0.1667 under methacholine-stimulated conditions. (D) Percentage of microdisks that reached the larynx during the study. p=0.1667 under basal conditions. * indicates p=0.0238 under methacholine-stimulated conditions.

challenges such as aspiration, irritants, and pathogens.

Another potential advantage of having SMGs is that the mucus they produce emerges onto the airway surface in the form of strands (*Hoegger et al., 2014*; *Ermund et al., 2018*; *Ostedgaard et al., 2017*; *Ermund et al., 2017*; *Fischer et al., 2019*; *Tipirneni et al., 2018*; *Trillo-Muyo et al., 2018*; *Xie et al., 2020*). Our ex vivo and in vivo data and earlier reports indicate that strands of mucus bind to large particles and transmit forces from beating cilia, thereby initiating and sustaining particle transport up the airways and out of the lung (*Hoegger et al., 2014*; *Fischer et al., 2019*). Consistent with these findings, disrupting mucus strands by breaking disulfide cross links between mucin molecules impairs MCT (*Fischer et al., 2019*). Thus, in addition to the abundance of mucus that SMGs produce, the unique architecture of mucus strands may be important for host defense in cartilaginous airways.

In pigs and humans, SMGs line the large cartilaginous airways, decrease in number as the airways become smaller, and are absent in very small airways. Thus, SMGs are present where the velocity and turbulence of air flow deposit most large particulate material, and they are missing in the small distal airways that large particles do not reach. Locating SMGs in larger airways positions the mucus strands they produce at sites where they can facilitate removal of impacted large particles. This relationship between airway size and the presence of SMGs holds across a variety of species, with a positive relationship between tracheal diameter and SMG volume (*Widdicombe and Wine, 2015*; *Choi et al., 2000*). In small mammals such as mice and rabbits, the lung's airways have few or no SMGs (*Widdicombe and Wine, 2015*; *Choi et al., 2000*; *Meyerholz et al., 2018a*; *Borthwick et al., 1999*). Large particles may be removed by the nose and/or rarely reach their intrapulmonary airways, thus obviating a need for SMGs and mucus strands. Interestingly, in the trachea of a large mammal, the horse, the number of SMGs is similar to that of man and other large mammals, but the volume of individual glands is smaller (*Widdicombe and Pecson, 2002*). Horses are obligate nose breathers, and their long, complex nasal turbinates may prevent most large particles from reaching the lung (*Widdicombe and Wine, 2015*). Thus, perhaps the volume of mucus strands required for effective MCT is reduced.

In addition to stimulated conditions, the data suggest that SMGs also contribute to host defense under basal conditions. For example, in the absence of cholinergic stimulation, *EDA-KO* pigs had decreased bacterial killing, decreased measures of MCT, and decreased ability to eradicate an

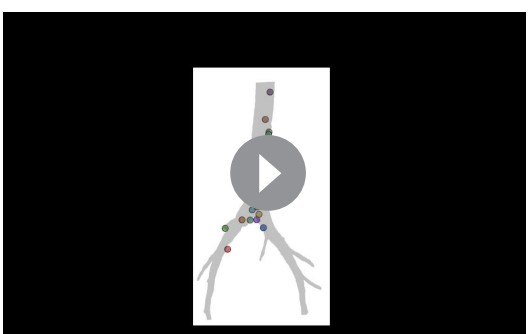

**Video 7.** Transport of microdisks in wild-type pig treated with IV methacholine. Procedures are as described in legend of *Video 5*. Pig is same as in *Video 5*.

https://elifesciences.org/articles/59653#video7

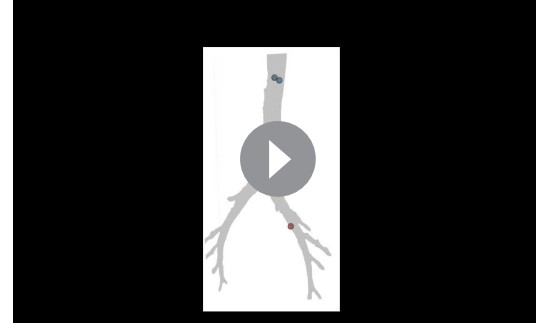

**Video 8.** Transport of microdisks in *EDA-KO* pig treated with IV methacholine. Procedures are as described in legend of *Video 5*. Pig is same as in *Video 6*.

https://elifesciences.org/articles/59653#video8

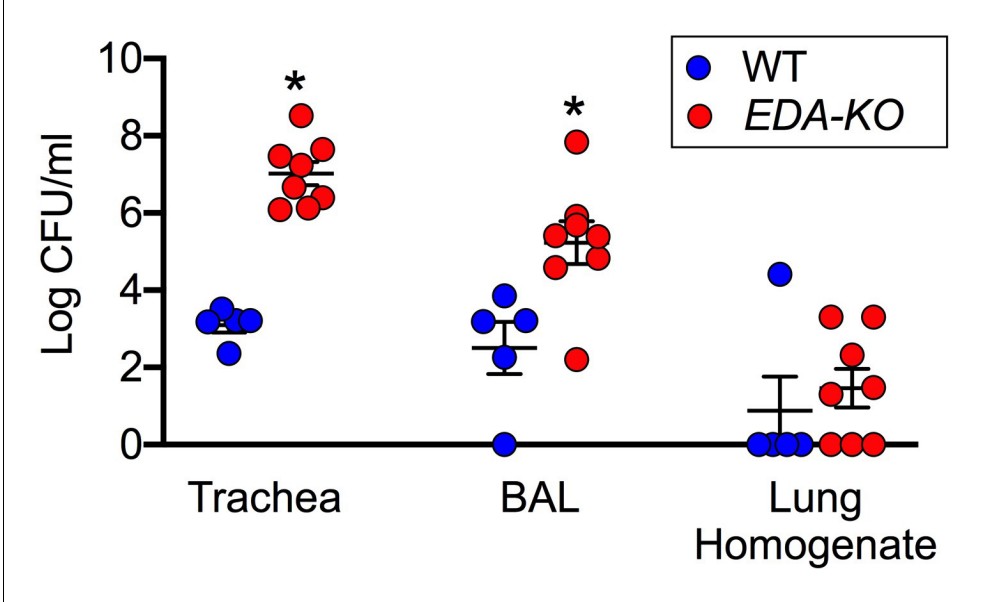

**Figure 8.** *EDA-KO* pigs have decreased eradication of *Staphylococcus aureus* from the lung. *S. aureus* were aerosolized into the airways and 4 hr later samples were obtained by tracheal washes, bronchoalveolar lavage (BAL), and lung homogenates. Data are the log colony-forming units recovered. N = 5 wild-type and 8 *EDA-KO* pigs. * indicates p=0.0016 for trachea washes, * indicates p=0.0186 for BAL, and p=0.3908 for distal lung homogenates. Statistical analysis was with a Mann-Whitney test.

inoculum of *S. aureus*, all in vivo. These results are consistent with previous studies indicating that SMGs produce small amounts of mucus under non-stimulated conditions (*Widdicombe and Wine, 2015*; *Wine and Joo, 2004*; *Quinton, 1979*; *Ueki et al., 1980*; *Joo et al., 2001*).

Our study also has limitations. First, to avoid potential confounding variables from infection, inflammation, and airway remodeling, we studied newborn *EDA-KO* pigs. The impairment of host defenses in these animals predicts that they will develop lung disease with time. Although the data are limited, reports from humans with HED also suggest that *EDA-KO* pigs will develop disease (*Dietz et al., 2013*; *Clarke et al., 1987*; *Reed et al., 1970*; *Callea et al., 2013*). Learning how the airways of *EDA-KO* pigs change with time and the compensatory adaptations they develop will allow a comparison to other diseases and thus improve understanding of host defense. Second, we do not identify each of the defense proteins and molecules produced by SMGs. Such information will further understanding of the role of SMGs in defending the airways. Third, given that SMG volume can increase several fold and mucus production is abundant in chronic obstructive pulmonary disease, some forms of asthma, and CF (*Fahy and Dickey, 2010*; *Stoltz et al., 2015*; *Widdicombe et al., 1994*; *Hogg, 2004*; *Hays and Fahy, 2006*; *Bonser and Erle, 2017*; *Ma et al., 2018*; *Turner and Jones, 2009*; *Boucherat et al., 2013*), it will be informative to know the contribution of SMGs to disease. Would such diseases be more or less severe without SMGs?

It has long been assumed that SMGs contribute to respiratory host defense (*Widdicombe and Wine, 2015*; *Wine and Joo, 2004*; *Ganz, 2002*; *Whitsett, 2018*; *Knowles and Boucher, 2002*; *Basbaum et al., 1990*; *Fahy and Dickey, 2010*; *Joo et al., 2015*; *Dajani et al., 2005*; *Bartlett et al., 2013*; *Fischer et al., 2009*). However, that hypothesis had not been directly tested. Our in vivo and ex vivo studies show that without SMGs, antimicrobial killing of bacteria is reduced, MCT is impaired, and eradication of bacteria from the lung is decreased. The results also emphasize the importance of mucus in the form of strands that facilitate MCT in cartilaginous airways. Thus, SMGs are critical for host defense in an animal model that has lungs like humans (*Rogers et al., 2008*).

# Materials and methods

## Key resources table

| Reagent type (species) or resource | Designation | Source or reference | Identifiers | Additional information |
|---|---|---|---|---|
| Antibody | Anti-MUC5B (rabbit polyclonal) | Santa Cruz | Cat# Sc-20119 | IF (1:2000) |
| Antibody | Anti-MUC5AC (mouse monoclonal) | Novus Biologicals | Cat # NBP2-15196 | IF (1:5000) |
| Antibody | anti-β-tubulinIV (mouse monoclonal) | Biogenex | Cat# Mu178-5UC | IF(1:300) |
| Chemical compound, drug | Methacholine (acetyl-β-methylcholine chloride) | Sigma | Cat# A2251 | |
| Chemical compound, drug | *CleanCap Cas9* mRNA | TriLink Biotechnologies | Cat # L-7606 | 20 ng/ul |
| Commercial assay or kit | MEGAshort script T7 Transcription kit | Thermo-fisher | Cat# AM1354 | |
| Commercial assay or kit | KAPA Express Extract Kit Plus amplification module | KAPA Biosystems | Cat# KK7152 | |
| Commercial assay or kit | MEGAclear Transcription Clean-up kit | Thermo-fisher | Cat# AM1908 | |
| Commercial assay or kit | QIAquick PCR Purification Kit | Qiagen | Cat# 28104 | |
| Commercial assay or kit | Live/Dead Bacterial Viability Assay | Thermofisher | Cat # L13152 | |
| Commercial assay or kit | TOPO TA Cloning kit (with PCR 2.1-TOPO vector) | Invitrogen | Cat# K45-0001 | |
| Gene (*Sus scrofa*) | EDA1 | Ensembl | Ensembl gene link: ENSSSCG00000021647 | |
| Other | Visualizing media | Invitrogen | 4 nm Nano spheres | 1:10000 |
| Other | Visualizing media | BalTec | Tantalum spheres | 500 µm |
| Other | Visualizing media | Sigma | Tantalum disks | 350 µm |
| Sequence-based reagent | EDA guide 1 | This paper | Guide RNA | GGAATCCCTG GAATCCCTGG |
| Sequence-based reagent | EDA guide 2 | This paper | Guide RNA | GCCCGGTGGT CCCATAACAG |
| Sequence-based reagent | Forward primer | This paper | Primer | gcctgactttgtgttg ttagaagtccata |
| Sequence-based reagent | Reverse primer | This paper | Primer | ctgctcttggtatca tgtactcctgatct |
| Software, algorithm | Imaging software | Olympus | CellSens | |
| Software, algorithm | Imaging software | NIH | ImageJ | |
| Software, algorithm | Imaging software | Nikon | NIS Elements | |
| Software, algorithm | Imaging software | Siemens | SOMATON Force | |
| Software, algorithm | Data analysis | GRAPHPAD Software | GRAPHPAD PRISM | |
| Software, algorithm | Data analysis | ITK-SNAP | ITK-SNAP | |
| Software, algorithm | Primer-Blast | NCBI | https://www.ncbi.nlm.nih.gov/tools/primer-blast/index.cgi | |

*Continued on next page*

*Continued*

| Reagent type (species) or resource | Designation | Source or reference | Identifiers | Additional information |
|---|---|---|---|---|
| Strain, strain background | *S. aureus* | *Pezzulo et al., 2012* PMID:22763554 | *S. aureus isolate 43SA* | |

## Generation and identification of EDA1-KO piglets

### Guide RNA design and preparation

Guide RNA sequences for two sgRNAs targeting exon 4 of porcine EDA1 were identified by using the Benchling (https://benchling.com/) and ChopChop v2 (*Labun et al., 2016*) web tools: Guide 1, 5'-GGAATCCCTGGAATCCCTGG-3'; Guide 2, 5'-GCCCGGTGGTCCCATAACAG-3'.

Guide RNAs (sgRNAs) were in vitro transcribed from gBlock gene fragments (Integrated DNA Technologies) that were synthesized to contain a T7 promoter sequence upstream of the sgRNA sequence as previously described (*Whitworth et al., 2017*). The guide RNAs were generated using the MEGAshortscript T7 Transcription Kit (Thermo Fisher) and purified using the MEGAclear Transcription Clean-Up Kit (Thermo Fisher). The concentration of the transcripts was determined using a Nanodrop spectrophotometer and the quality of the transcripts was analyzed by visualization on a 5.0% Criterion TBE-urea polyacrylamide gel (BIO-RAD).

### Production of pigs by injecting zygotes with Cas9/sgRNA

Both guides were mixed together in water with capped and polyadenylated *s.p.Cas9* mRNA (Trilink Biotechnologies) and the mixture containing 10 ng/µl of each guide and 20 ng/µl *Cas9* mRNA was injected into the cytoplasm of in vitro derived porcine zygotes. Embryos were then cultured for 5 days in MU2 (early reps) or MU3 (later reps) supplemented with FLI (FLI 40 ng/mL FGF2, 20 ng/mL LIF, 20 ng/mL IGF) (*Yuan et al., 2017*; *Chen et al., 2018*; *Redel et al., 2019*). Blastocyst-stage embryos were then surgically implanted into the oviduct of surrogate sows.

### Piglet genotyping assay

Tail or ear tissue collected from the piglets was lysed using the KAPA Express Extract Kit Plus Amplification Module (KAPA Biosystems). Module and targeted fragments were amplified from the crude tissue lysates using KAPA 2G polymerase (KAPA Biosystems). The primers for amplifying the targeted fragments were designed using the Primer-Blast Program (*Ye et al., 2012*): Forward Primer, 5'-GGC TGA CTT TGT GTT GTT AGA AGT CCA TA-3'; Reverse Primer, 5'-CTG CTC TTG GTA TCA TGT ACT CCT GAT CT-3'. PCR conditions consisted of an initial denaturation of 95℃ for 3 min, followed by 40 cycles of 95℃ (15 s), 60℃ (15 s), 72℃ (15 s), and a final extension cycle of 72℃, (1 min).

The PCR products were purified using the QIAquick PCR Purification Kit (Qiagen) and cloned into the TOPO TA vector, PCR 2.1-TOPO, and transfected into TOPO One Shot cells. Kanamycin-resistant colonies were picked and sequenced for analysis of indels (Functional Biosciences).

All piglets from *EDA-KO* litters were edited. Wild-type controls were from other litters at the University of Missouri or were purchased from Exemplar Genetics. Animals were sedated with ketamine/xylazine (Akorn), and sedation was maintained with propofol. Euthanasia was with Euthasol (Virbac) after ketamine sedation. The Animal Care and Use Committees (IACUC) at the University of Iowa and the University of Missouri approved all animal studies conducted at their respective locations.

## Histopathological analyses

Tissues were placed in 10% neutral buffered formalin (5–7 days), dehydrated through a series of alcohol and xylene baths, paraffin-embedded, sectioned (4 µm) and stained with hematoxylin and eosin (HE) or diastase-pretreated Periodic acid Schiff (dPAS) (*Meyerholz et al., 2018b*). Tissues were examined using the post-examination method of masking and scored following principles for reproducible histopathologic scores (*Meyerholz and Beck, 2018c*). Submucosal glands were evaluated in trachea, secondary bronchi, segmental bronchi and bronchioles. In each airway, the length of airway luminal circumference with subjacent submucosal gland cells was divided by the total circumference of the airway to produce a '% circumference with SMG'.

## Immunocytochemistry

Trachea were excised from newborn piglets and immediately fixed in 4% paraformaldehyde (EMS) in PBS for 1 hr at room temperature. Tissues were then placed in 30% sucrose and incubated overnight at 4°C, followed by quick-freezing in OCT using a dry ice/EtOH bath and stored at −80°C. Prior to immunocytochemistry, frozen blocks of tissue were cryosectioned at 7 µm followed by permeabilization in 0.3% TX-100 (Thermo-Fisher) in PBS for 20 min, and blocked in Super-Block (Thermo-Fisher) with 5% normal goat serum (Jackson ImmunoResearch) for 1 hr, all at room temperature. Tissue sections were then incubated for 2 hr at 37°C with indicated antibodies: β-tubulin IV(1:300, Biogenex), MUC5AC (1:5000, Novus Biologicals), MUC5B (1:2000, Santa Cruz). Sections were then incubated for 1 hr with secondary antibodies goat-anti-mouse Alexa-Fluor-488 and goat anti-rabbit Alexa-Fluor-555 (1:1000, Molecular Probes/Invitrogen) and phalloidin-633 (1:300, Molecular Probes/Invitrogen). Slides were imaged on an Olympus Fluoview FV3000 confocal microscope with a Plan.ApoN 60X oil lens. Images were post-processed using the Olympus imaging software, CellSens.

## Assay of bacterial killing

Antimicrobial activity measurements were performed using bacteria-coated grids. Preparation, imaging, and quantification was performed as previously described (*Pezzulo et al., 2012*). *S. aureus* isolate 43SA was cultured to log-phase growth, labeled with biotin, and conjugated to gold electron microscopy grids coated with streptavidin. The bacteria-coated grids were placed on the airway surface through a tracheal window of a sedated pig for 1 min, rinsed with PBS, and immersed in SYTO9 and propidium iodide (Invitrogen) to determine bacterial viability (Live/Dead Bacterial Viability Assay, Invitrogen). Two technical replicates were performed per pig and the results averaged. Numbers of live and dead bacteria on grids were analyzed with confocal microscopy and quantified by Image J (ImageJ, Schneider, CA, USA).

## In vitro assay of MCT

### Tissue preparation

Trachea explants were pinned to dental wax and submerged in 40 ml of Krebs buffered saline at pH 7.4 in 5% $CO_2$ in a 37°C chamber (*Hoegger et al., 2014*; *Fischer et al., 2019*). Ciliary beat frequency, strand counting, and metallic sphere transport experiments shown in this study were all performed after treatment with 100 µM methacholine.

### Ciliary beat frequency

Ciliary beating was visualized as previously described (*Hoegger et al., 2014*; *Fischer et al., 2019*). The epithelial surface was imaged using reflected light with a Nikon A1 confocal microscope with 25X submersion lens. Video recordings were obtained at 110 frames per second for 4 s using NIS elements software. Three separate microscopy fields were collected per animal. Two observers independently calculated the ciliary beat frequency by importing image stacks into FIJI, drawing polygons around ciliated cells, and measuring the frequency of oscillation in the reflected light channel. Correlation between observers was >0.95. Each data point represents the average of all fields made by both observers.

### Count of moving strands

Strand counting was measured as previously described (*Fischer et al., 2019*). The epithelial surface was imaged using reflected light and mucus strands were visualized by 1:10,000 addition of 4 nm fluorescent nanospheres (Invitrogen). The field was recorded for 15 min at a frame rate of 8–10 frames/sec. Two observers independently calculated the number of strands crossing the field by first drawing a line perpendicular to the direction of mucociliary transport then marking two points separated by 140 µm on that line. Any continuous mucus strand that crossed the field and touched both pre-defined points was counted. Correlation between observers for strand count was >0.95. Each data point represents the average of the number of strands counted by both observers.

### Transport of metallic spheres

Ta spheres (500 µm diameter, Bal-tec, Los Angeles) were added to the tracheal surface. We monitored their transport for 15 min by time lapse photography using the Frameography application for

iPhone as previously described (*Hoegger et al., 2014*; *Fischer et al., 2019*). Because we previously observed that disruption of mucus impairs the initiation of transport, we measured the fraction of spheres that moved >1 mm from their original position during the tracking period. To determine the role of mucus strands in transporting Ta spheres, we added 4 nm fluorescent nanospheres and single Ta spheres simultaneously. We visualized the interaction of the sphere with mucus using a Nikon A1 confocal microscope with 10X lens. The reflected light channel was used to visualize spheres and a red fluorescence channel to visualize mucus.

## In vivo MCT assay

### X-ray computed tomographic (CT) assay

We sedated animals for in vivo studies with ketamine (20 mg/kg, I.M., Phoenix Pharmaceutical, Inc) and acepromazine (2 mg/kg, I.M., Phoenix Pharmaceutical, Inc) or xylazine (2 mg/kg, I.M., Lloyd). Anesthesia was maintained with I.V. dexmedetomidine (10 µg/kg/hr, I.V., Accord Healthcare, Inc).

To measure MCT in vivo, we used a previously described CT-based assay (*Hoegger et al., 2014*; *Fischer et al., 2019*). We measured MCT by tracking tantalum microdisks (350 µm diameter x 25 µm thick, Sigma). To deliver microdisks, animals were anesthetized, briefly intubated, and microdisks were insufflated into the airways just beyond the vocal cords with a puff of air. Immediately after delivery, the tubes and catheter were removed. CT scans were acquired with a continuous spiral mode CT scan (0.32 s rotation; 176 mm coverage in 1.5 s; 0.6 mm thick sections with 0.3 mm slice overlap, Siemens SOMATOM Force). Forty-four CT scans were obtained in a 6.3 min time interval. Microdisks were tracked over time by an automated algorithm (FiJi TrackMate plugin [*Tinevez et al., 2017*]) and manually validated.

### CT scan data analysis

Microdisks that failed to move >10 mm from their initial position were labeled as non-moving. We used that number to calculate the average % of time microdisks were in motion for each individual pig. Tracking microdisks over time provided multiple measurements of microdisk speed. From these speeds, we determined the average speed of individual microdisks after they had moved >10 mm and used that to determine the mean speed for all the microdisks in each pig. Microdisk clearance was calculated by determining whether a microdisk reached the larynx or not during the 6.3-min tracking period. The percentage of microdisks cleared was determined by dividing the number of cleared microdisks by the total number of microdisks tracked x100%. The airway tree from each animal was segmented using thresholding segmentation mode in ITK-SNAP with an upper limit of −600 HU (*Yushkevich et al., 2006*). An anteroposterior projection of the segmented airway tree mesh was overplayed on top of each video.

## Bacterial challenge

Two-day-old *EDA-KO* (n = 8) and wild-type (n = 5) pigs received an intrapulmonary challenge with log-phase chloramphenicol-resistant *S. aureus* (average inoculum 1.0–1.4 $\times$ 10$^8$ CFU) delivered in 0.1 ml of 0.45% saline using a MADgic atomizer positioned just distal to the vocal cords. Four hours later, pigs were euthanized and the trachea was removed and divided in upper and lower halves. Each tracheal piece was washed with 1 mL of PBS containing $Ca^{2+}$ and $Mg^{2+}$ and the solution was vortexed for 30 s. Bronchoalveolar lavage was performed on the right and left lower lobes (5 ml of saline on each side), repeated three times and pooled and the numbers of bacteria for each pig were averaged. The right accessory lobe was homogenized with manual tissue grinders in 3 ml of PBS containing $Ca^{2+}$ and $Mg^{2+}$, spun for 1 min on a tabletop minicentrifuge, and the pellet was discarded. CFU/ml were quantitated from 10-fold serial dilutions of each recovered sample cultured on TSB plates containing chloramphenicol.

## Statistical analysis

Data are presented for individual animals with mean ± SEM. Statistical comparisons between wild-type and *EDA-KO* were by a Mann-Whitney test. Differences were considered statistically significant at $p < 0.05$. Analyses were made in GraphPad Prism v7.0d (GraphPad Software, La Jolla, CA).

## Acknowledgements

We thank the Comparative Pathology Laboratory, the Central Microscopy Research Facility, and the Genomics Division of the Iowa Institute of Human Genetics for technical assistance. We thank Jason Dowell at the University of Missouri Swine Research Complex and Elizabeth Blackstock for their management of the surgeries and care of the surrogates. We thank Raissa Cecil, Caroline Pfeiffer, Bethany Redel, Joshua Benne and Taylor Hord who performed the zygote injections for these experiments. We thank Harlee Brommel, Brooklyn Freas, Joseph Stoltz, Alexis Hansen, Ana De La Torre, Frank Iole, and Bradley Boysen at the University of Iowa for assistance with the animals and for particle tracking. This work was supported, in part, by the National Institutes of Health (NIH) and Cystic Fibrosis Foundation (CFF): NIH K08 HL136927 and CFF FISCHE16I0 to AJF, NIH K08 HL135433 to MAA, CFF STOLTZ16X $\times$ 0 to DAS and MAA, NIH R01 HL136813 to DAS, NIH-S10OD018526 to EAH, NIH PPG (HL091842, HL051670) (DAS and MJW), and a CFF Research Development Program. MJW is an investigator of the Howard Hughes Medical Institute.

## Additional information

### Funding

| Funder | Grant reference number | Author |
|---|---|---|
| National Institutes of Health | NIH K08 HL136927 | Anthony J Fischer |
| Cystic Fibrosis Foundation | CFF FISCHE16I0 | Anthony J Fischer |
| National Institutes of Health | NIH K08 HL135433 | Mahmoud Abou Alaiwa |
| Cystic Fibrosis Foundation | CFF STOLTZ16XX0 | Mahmoud Abou Alaiwa David Stoltz |
| National Institutes of Health | NIH R01 HL136813 | David Stoltz |
| National Institutes of Health | NIH-S10OD018526 | Eric A Hoffman |
| National Institutes of Health | NIH PPG HL091842 | David Stoltz Michael J Welsh |
| National Institutes of Health | NIH PPG HL051670 | David Stoltz Michael J Welsh |
| Cystic Fibrosis Foundation | Research Development Program | David Stoltz Michael J Welsh |
| Howard Hughes Medical Institute | Investigator | Michael J Welsh |

The funders had no role in study design, data collection and interpretation, or the decision to submit the work for publication.

### Author contributions

Lynda S Ostedgaard, Conceptualization, Data curation, Formal analysis, Supervision, Investigation, Methodology, Writing - original draft, Project administration, Writing - review and editing; Margaret P Price, Kristin M Whitworth, Resources, Data curation, Formal analysis, Investigation, Methodology, Writing - review and editing; Mahmoud H Abou Alaiwa, Anthony J Fischer, Resources, Data curation, Formal analysis, Investigation, Visualization, Methodology, Writing - review and editing; Akshaya Warrier, Melissa Samuel, Lee D Spate, Brieanna M Hilkin, Guillermo S Romano Ibarra, Miguel E Ortiz Bezara, Brian J Goodell, Steven E Mather, Linda S Powers, Mallory R Stroik, Nicholas D Gansemer, Camilla E Hippee, Keyan Zarei, Data curation, Formal analysis, Investigation, Methodology, Writing - review and editing; Patrick D Allen, Data curation, Formal analysis, Methodology, Project administration, Writing - review and editing; J Adam Goeken, Thomas R Businga, Data curation, Investigation, Methodology, Writing - review and editing; Eric A Hoffman, Supervision, Funding acquisition, Methodology, Writing - review and editing; David K Meyerholz, Randall S Prather, Resources, Data curation, Formal analysis, Supervision, Funding acquisition, Investigation, Methodology, Writing - review and editing; David A Stoltz, Michael J Welsh, Conceptualization, Resources, Data curation, Formal

analysis, Supervision, Funding acquisition, Investigation, Methodology, Writing - original draft, Project administration, Writing - review and editing

### Author ORCIDs
Miguel E Ortiz Bezara http://orcid.org/0000-0002-3573-5416
Eric A Hoffman http://orcid.org/0000-0001-8456-9437
David K Meyerholz https://orcid.org/0000-0003-1552-3253
Michael J Welsh https://orcid.org/0000-0002-1646-6206

### Ethics
Animal experimentation: The Animal Care and Use Committees (IACUC) at the University of Iowa (Protocol #7071121) and the University of Missouri (Protocol #9643) approved all animal studies conducted at their respective locations.

### Decision letter and Author response
Decision letter https://doi.org/10.7554/eLife.59653.sa1
Author response https://doi.org/10.7554/eLife.59653.sa2

## Additional files

### Supplementary files
• Transparent reporting form

### Data availability
All data generated or analysed during this study are included in the manuscript and supporting files.

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
