## [Decision Letter]

**Acceptance summary:**

We believe that your study on the role of submucosal glands in host immunity will add critical insight into how this airway niche regulates the response to infectious diseases.

**Decision letter after peer review:**

Thank you for submitting your article "Lack of Airway Submucosal Glands Impairs Respiratory Host Defenses" for consideration by *eLife*. Your article has been reviewed by two peer reviewers, and the evaluation has been overseen by a Reviewing Editor and Edward Morrisey as the Senior Editor. The reviewers have opted to remain anonymous.

The reviewers have discussed the reviews with one another and the Reviewing Editor has drafted this decision to help you prepare a revised submission.

The current study addresses a scientific question of high interest and further substantiate the long-standing assumption that SMG s are critical for muco-ciliary clearance. The study is well performed with many robust results, which support the conclusions and warrants publication in *eLife*.

These following points were raised by the reviewers, which primarily requires revisions of the manuscript text and Discussion. These points need to be addressed by the authors before acceptance.

1) The authors overstate the novelty of the work and do not fully acknowledge the very large body of evidence that supports the role of submucosal glands on airway defence. The manuscript under review is not the first research studying the effect of inhibiting submucosal gland on airway. A quick search in Pubmed for Airway Submucosal Glands renders more than 700 publications. Many of them using pharmacology and other interventions to model the lack of submucosal glands. For example, the authors of the manuscript under review published a paper where they studied tracheal xenograft with and without glands to study the contribution of submucosal glands to antimicrobial properties of the airway. Please adapt the Introduction and Discussion section of the manuscript accordingly.

2) Another aspect of this manuscript that is notable is that the results match what we would have expected based on the literature, as summarized by the authors in a review 5 years ago (Stoltz, Meyerholz and Welsh, 2015). Thus, the results in this manuscript are confirmatory of previous hypotheses.

3) The authors state: "Moreover, the assumption that SMGs are important for respiratory host defense is challenged by the observation that small mammals such as mice and rabbits lack abundant SMGs and have only a few SMGs near the larynx (Widdicombe and Wine, 2015; Choi, Finkbeiner and Widdicombe, 2000; Meyerholz et al., 2018; Berthwick et al., 1999)." The lack of submucosal glands in rabbits and mice does not necessarily challenge the hypothesis that glands play a role in host defense in animals that do have them (e.g. humans and swine). In the past it has been proposed that the lack of glands in rabbits and mice may be explained by the observation that these animals are obligate nose breathers and have a different nasal anatomy than swine or human. In fact, the paper cited by the authors as #1 has a very different interpretation than that provided in the current manuscript. That reference states: "Rabbit and horse trachea have fewer glands than expected for other animals of their size, presumably because they are obligate nose breathers with unusually long and convoluted nasal turbinates that contain abundant submucosal glands…."

4) Subsection “DA-KO piglets lacked airway SMGs”, second paragraph. Submucosal glands are a major source of MUC5B. Do *EDA-KO* swine have less MUC5B staining than WT?

5) Figure 7B and subsection “*EDA-KO* piglets had impaired MCT in vivo”, third paragraph. The data *EDA-KO* swine data shows that one swine has tantalum disk movement that is similar to wild type. How do the authors explain that?

6) Subsection “Airways of *EDA-KO* piglets had an impaired ability to eradicate bacteria”, last paragraph. The *EDA-KO* swine had many more times *S. aereous* in the trachea, compared with WT. But there was no difference in the distal airway (homogenized sample of the lung). How is that possible? Wouldn't be expected that if the trachea of *EDA-KO* swine has 10,000 times more bacteria than WT, there should be a lot more bacteria reaching the distal airway?

---

## [Author Response]

[…] These following points were raised by the reviewers, which primarily requires revisions of the manuscript text and Discussion. These points need to be addressed by the authors before acceptance.1) The authors overstate the novelty of the work and do not fully acknowledge the very large body of evidence that supports the role of submucosal glands on airway defence. The manuscript under review is not the first research studying the effect of inhibiting submucosal gland on airway. A quick search in Pubmed for Airway Submucosal Glands renders more than 700 publications. Many of them using pharmacology and other interventions to model the lack of submucosal glands. For example, the authors of the manuscript under review published a paper where they studied tracheal xenograft with and without glands to study the contribution of submucosal glands to antimicrobial properties of the airway. Please adapt the Introduction and Discussion section of the manuscript accordingly.

We agree that much evidence suggests that SMGs play a role in airway defense. Accordingly, in the first paragraph we said, “it has been assumed that SMGs play an important role in respiratory host defense” followed by 9 references (the first 5 are reviews) including Dajani et al., 2005, to which the reviewer refers. We also agree that additional discussion and references can be provided; therefore, we have further expanded our discussion of earlier work in the fourth paragraph of the Introduction section. We do not repeat this in the Discussion section, but there we do refer to previous papers on the topic.

2) Another aspect of this manuscript that is notable is that the results match what we would have expected based on the literature, as summarized by the authors in a review 5 years ago (Stoltz, Meyerholz and Welsh, 2015). Thus, the results in this manuscript are confirmatory of previous hypotheses.

We agree completely, and now explicitly say that in the first paragraph of the Discussion.

3) The authors state: "Moreover, the assumption that SMGs are important for respiratory host defense is challenged by the observation that small mammals such as mice and rabbits lack abundant SMGs and have only a few SMGs near the larynx (Widdicombe and Wine, 2015; Choi, Finkbeiner and Widdicombe, 2000; Meyerholz et al., 2018; Berthwick et al., 1999)." The lack of submucosal glands in rabbits and mice does not necessarily challenge the hypothesis that glands play a role in host defense in animals that do have them (e.g. humans and swine). In the past it has been proposed that the lack of glands in rabbits and mice may be explained by the observation that these animals are obligate nose breathers and have a different nasal anatomy than swine or human. In fact, the paper cited by the authors as #1 has a very different interpretation than that provided in the current manuscript. That reference states: "Rabbit and horse trachea have fewer glands than expected for other animals of their size, presumably because they are obligate nose breathers with unusually long and convoluted nasal turbinates that contain abundant submucosal glands…."

We removed the statement to which the reviewer refers. Widdicombe and Wine, 2015, is an outstanding review of SMGs, and the discussion about species differences is interesting. However, the reference to Widdicombe and Wine, 2015, has an error in the quotation to which the reviewer refers; horses do not have fewer glands than expected. The primary paper by Widdicombe and Pecson (Widdicombe and Pecson, 2002) says, “The small volume of glands in the horse trachea is due to smaller size of individual glands rather than a reduction in the total number of glands. Thus, the numbers of openings in the ventral aspect of the horse trachea (1.0/mm^2^) is similar to that in both the ox (1.5/mm^2^) and man (1/mm^2^). an average volume per gland of 17 nl is obtained. By contrast, the volume of individual glands in other large mammalian species is ~120 nl (Choi et al., 2000)”. We have added a short discussion in the fourth paragraph of the Discussion.

4) Subsection “DA-KO piglets lacked airway SMGs”, second paragraph. Submucosal glands are a major source of MUC5B. Do EDA-KO swine have less MUC5B staining than WT?

*EDA-KO* lungs lack submucosal glands and hence the MUC5B expressed there. In surface epithelia, immunostaining revealed no genotype-dependent difference in MUC5B or MUC5AC localization. In the future, it will be important to obtain quantitative information about mucins in surface epithelia and in the airway lumen.

5) Figure 7B and subsection “EDA-KO piglets had impaired MCT in vivo”, third paragraph. The data EDA-KO swine data shows that one swine has tantalum disk movement that is similar to wild type. How do the authors explain that?

For one of the *EDA-KO* pigs, the % of time disks were in motion approached that of wild-type pigs after administration of methacholine (Figure 7B). Despite this, MCT was not normal as indicated by: (a) the % of time disks were in motion was less than that of wild-type pigs under basal conditions (Figure 7B), (b) the speed of microdisks that did move tended to be slower in *EDA-KO* pigs (Figure 7C), and (c) none of the disks cleared from the lungs of *EDA-KO* pigs (Figure 7D). That said, we do not know why there is animal to animal variability, but it may be worth noting that the pigs were not inbred.

6) Subsection “Airways of EDA-KO piglets had an impaired ability to eradicate bacteria”, last paragraph. The EDA-KO swine had many more times S. aereous in the trachea, compared with WT. But there was no difference in the distal airway (homogenized sample of the lung). How is that possible? Wouldn't be expected that if the trachea of EDA-KO swine has 10,000 times more bacteria than WT, there should be a lot more bacteria reaching the distal airway?

There are several considerations. First, because submucosal glands reside in more proximal airways, we expect that their loss will have a greater impact on antimicrobial-dependent bacterial killing and MCT of bacteria in those airways where they are located. In contrast, we expect that antimicrobial activity and MCT may be less affected in distal parts of *EDA-KO* lungs. Second, the quantitative delivery of *S. aureus* to various airways is not known. We suspect that delivery is greater to more proximal airways, but we cannot be certain. Although CFUs tend to be larger in tracheal washes than in lung homogenates of wild-type pigs, we do not know the dilution of ASL in the three sampling procedures, and bacterial killing and clearance may differ proximally and distally. We now address these points in the last paragraph of the subsection “Airways of *EDA-KO* piglets had an impaired ability to eradicate bacteria”.